# Fertility Practitioners' Coping Strategies When Faced with Intra-Role Conflict from Screening Aspiring Single Mothers by Choice

**Paulien Hertogs** [1] **, Dries Van Gasse** [1,*] **, Sascha Spikic** [1,2] **and Dimitri Mortelmans** [1]

1 Center for Population, Family and Health, Faculty of Social Sciences, University of Antwerp, 2000 Antwerpen, Belgium; paulien.hertogs.2@gmail.com (P.H.); Sascha.Spikic@uantwerpen.be (S.S.); Dimitri.Mortelmans@uantwerpen.be (D.M.)
2 Faculty of Engineering Sciences, KU Leuven, 3000 Leuven, Belgium
* Correspondence: Dries.vangasse@uantwerpen.be

**Abstract:** Women without a partner can become single mothers by choice through the use of fertility treatments. In Belgium, the decision to accept a candidate single mother by choice rests with the fertility clinic's multidisciplinary team of fertility practitioners. As a result, the fertility practitioners fulfil a gatekeeping role. However, this can cause an intra-role conflict as the responsibility to select the best fitting candidates is at odds with the responsibility to help patients. In this explorative study, we examine how fertility practitioners cope with the strain resulting from intra-role conflict in the decision-making process regarding single motherhood by choice in Belgium. The findings showed that practitioners appear to mainly resort to problem-focused coping, by constructing a grassroots criteria list and by shifting their role from screening agent to counsellor. These results are based on ten open in-depth interviews with fertility practitioners employed in the multidisciplinary teams of fertility centers, using a reflexive interview lead.

**Keywords:** (clinical) gatekeeping; fertility; single mother by choice (SMC); solo mother

## 1. Introduction

> "We don't need an additive legislation as such, but we need scientific grounding. We have to address the question 'What is a good parent?', but yeah, what does make a parent a good parent?". (Psychologist 6)

In many parts of Western Europe and North America, the concept of family has moved "beyond the nuclear ideal" (Cutas and Chan 2012). This diversification of the concept of family is not the result of any single trend, but of several, including the normalization of divorce, a decline of heteronormativity in relations, and the conception of single motherhood by choice via ART (Assisted Reproductive Technology) (Bernardi et al. 2018; Hertz 2006; and Hopkins et al. 2013). The expanding interpretation of what constitutes a family has left policymakers struggling to keep up and, consequently, family policies are not always adequately tailored to the currently existing and emerging family constellations (Dermott and Fowler 2020). Many family forms now deviate from the so-called traditional heterosexual two-parent families that social policies target. In this study, we focus on the case of single motherhood by choice: the dyad of mother and child that constitutes a family system on its own, planned in a prospective manner (Hertz 2006). Despite the emergent possibility of becoming a parent on one's own, previous studies have documented that the occurrence of this family type in the United States, the United Kingdom, Spain, and Belgium should not be perceived as a free passage (Hertz et al. 2016; Golombok et al. 2016; Murray and Golombok 2005; and Zadeh et al. 2013).

Single motherhood by choice is a unique family type, as clinical fertilization has enabled aspiring single women to start families without partners—which was unthinkable in

the past—through the use of a donor sperm (Hertz 2006). The option of clinical fertilization stands out because it allows single women to conceive (in contrast with adoption) without the physical presence of a male donor (Baldwin 2019). The earliest adopters of these medical possibilities did so without the existence of any legal framework (Graham 2012). Since then, some countries have implemented a legal framework regarding the admittance of aspirant mothers to assisted fertility programs. In Belgium, the country in which this study took place, the legal framework regarding donor fertilization was formed in 2007. It grants clinical practitioners a high level of agency (Pennings 2012) since there are few guidelines regarding the screening procedure and much reliance on a moral clause for the clinical practitioners to reject aspiring single mothers based on ethical considerations that would not meet the best interest of the child. In this respect, we assume the best interest of the patient is to have their wish to become a single mother by choice be fulfilled, whereas the best interest of the child is to be born in a well-functioning family unit. Therefore, the moral clause effectively puts the responsibility on the clinical practitioners and thus assigns to them a gatekeeping role as they are tasked with evaluating if the candidates are able to manage the role of single mother. We will refer to the multidisciplinary team who take these decisions as fertility practitioners. This term is preferred to fertility doctor or fertility specialist, because the multidisciplinary teams also includes members without a medical background, such as psychologists.

Single mothers by choice defy the social norm of coupled parenthood. Fertility practitioners themselves are also influenced by these social norms, which makes the decision more complex. After all, the social conceptualization of good parenting is formed by the interaction of different social views (Morris et al. 2020). It is plausible that these social norms steer fertility practitioners, but also an "objective" assessment leads to a rejection of potential single parents with low financial and/or social capital because it is believed that these characteristics aggravate the challenges of single parenthood (Whisenhunt et al. 2019). In short, social norms and values may lead to a biased selection of single mothers by choice (Golombok et al. 2016).

In their gatekeeping role, fertility practitioners face conflicting responsibilities as well. Rejected candidates can also become a parent via other means, through which they receive less formal support in the preparation of parenthood. This puts pressure on practitioners because rejection might not withhold aspiring single parents to become a parent. When fertility practitioners reject candidate single parents, they cannot help them in their transition process as well. This intra-role conflict can lead to feelings of role strain (Dasgupta and Kumar 2009). However, not much is known about the coping strategies fertility practitioners implement to help alleviate their feelings of role strain. Hence, our research question is: *how do fertility practitioners cope with the strain resulting from intra-role conflict in the decision-making process of single motherhood by choice in Belgium*? To answer this question, we examined the attitudes of practitioners in fertility clinics towards prospective single mothers by choice with the aim of clarifying the decision-making process. In particular, we focused on the concerns arising from the current legal framework and how clinical practitioners deal with these concerns.

## 2. Literature

### 2.1. The Belgian Context

This study took place in Belgium, a country we will first situate and contextualize from a cultural perspective, as this can have an impact on the dominant parenthood ideology. Thereafter, we describe the practice in fertility clinics. From a religious perspective—which can affect the norms and values on parenthood, reproduction, etc.—Belgium has a Catholic tradition, which is demonstrated by the fact that 57.8% of the population still affiliates with the Roman Catholic Church (Sealy and Tariq 2019). Although increased immigration rates since the 1970s have spurred cultural diversity, the second and third biggest religious affiliations in Belgium are agnostics (20.2%) and atheists (9.1%). The Catholic tradition is also partially embedded in Belgian politics, with the presence of a Christian party that had a

vast prominence in the political landscape. Nonetheless, the strong Catholic tradition of this party has shifted in recent decades towards a more humanist identification and discourse (Koutroubas et al. 2011). Due to its multicultural and multilinguistic diversity, Belgium has a complex socio-political structure with three territorial communities—Brussels, Wallonia, and Flanders—and three linguistic communities—French, Dutch, and German (Pew 2017).

The first law on medically assisted procreation in Belgium passed in 1999. The successive construction of a legal framework regarding assisted reproduction took place between 1999 and 2007, ending with the "law on medical assisted procreation and the destination of superfluous embryos and gametes," regulating the substantial practices in fertility clinics (Pennings 2007). In a European context, the integration of regulations concerning medically assisted procreation in Belgium happened rather late (Schiffino and Varone 2004). Moreover, Schiffino and Varone (2004) described Belgium as a bioethical paradise. In everyday Belgian practice, a laisser-faire policy hands all agency of fertility gatekeeping to the multidisciplinary teams of fertility clinics. Any regulation limiting fertility treatments to heterosexual infertile couples may be regarded as discriminative; therefore, various countries forgo specific requirements regarding assisted reproduction, aside from an age limit (Nordqvist 2012).

### 2.2. Single Mothers by Choice Challenging the Conception of Parenthood in the 21st Century

Single motherhood by choice is an alternative way of starting a family (Hertz 2004). Hertz (2006) described single mothers by choice as women finding themselves at an intersection of biological and social pressures without having a partner to fulfil their parental aspirations. Despite the fact that there are no available statistics on single parents by choice, it is clear that they emerged in the past decades and might become more prevalent in the future (Hayford and Guzzo 2015). As Passet-Wittig and Greil (2021) argue, the existing research on their prevalence often fails to map single parenthood by choice properly due to conceptual differences and single-country focuses, but public awareness has grown much in the recent decades. Similar to traditional parenthood, single parenthood by choice constitutes the normative transition to the parenting stage in the life course in which people acquire a new social role (Mccubbin and Figley 2014). As such, the transition to single motherhood by choice can be perceived as a quintessential integrative event into the peer community of parents (Rossi 1968). By starting a family on their own, single parents by choice move beyond earlier conceptions of parenthood and challenge several social assumptions about the constitution of families. In this specific case of parenting, parenthood is enabled by fertility practitioners. Since only a couple of studies focus on gatekeeping (with some exceptions, e.g., Johnson 2012), we contribute by expanding the existing literature to the perspective of fertility practitioners. This is important since, in their gatekeeping role, fertility practitioners vocalize important norms and values about parenting in society.

Fertility gatekeeping defines the norms and values of parenthood in the 21st century, because it expresses dominant parenthood ideologies in a very applied manner (Sperling and Simon 2010). It is widely documented that there exist implicit cultural definitions about parenthood that are performative in the everyday behavior of parents (Marsiglio et al. 2000; Purewal and van Den Akker 2007). It has also been shown that the performativity of these dominant parenthood ideologies become visible in specific cases—such as in the division of unpaid and paid work (Glauber and Gozjolko 2011), the use of parental leave (Van Gasse et al. 2021), or even family leisure activities (Shaw 2008)—but in these cases, parenthood ideologies always work in an implicit manner: social expectations based on intersectional identities nudge parents towards a certain behavior (Taylor 2011). In contrast with those other areas, a parenthood ideology is explicitly voiced when single women are assessed on parental fitness. Fertility gatekeeping is an often unsolicited social role practitioners in fertility clinics have to take on. Fertility gatekeeping also means that not each and every patient can be treated, although this is often voiced as a main motivation for students pursue careers in the healthcare sector (Newton et al. 2009). Instead, the moral clause

and related need for selection changes the role of practitioners into judges that deny some candidates entrance into the fertility procedure and, hence, the transition to parenthood. In the next paragraphs, we will discuss how this can lead towards role conflicts more in depth.

### 2.3. Intra-Role Conflict

Involving practitioners as fertility gatekeepers can result in a role conflict. Parsons (2003) defined a role conflict as "the exposure of an actor to conflicting sets of role expectations such that complete fulfilment of both is realistically impossible". As Kühne and Leonardi (2020) argued, there is a difference between inter-role conflicts, when people take up different conflicting social roles, and intra-role conflicts, when conflict arises due to competing interests of the same role. By appointing fertility practitioners as gatekeepers, Storrow (2006) argues that these gatekeepers attain a responsibility towards the unborn child. Practitioners thus not only have a care responsibility towards their patients, but should also safeguard the well-being of the child.

### 2.4. Biases and Discrimination

The consequent selection of single parents by choice shows a rather homogeneous picture. According to Golombok et al. (2016) and Richards et al. (2012), single mothers by choice are often in their late 30s or early 40s, highly educated, and financially independent. Jadva et al. (2009) added that these women usually have experienced long-term relationships in their pasts. According to Hertz (2006), this profile does not match the characteristics of the overall population of aspiring single parents by choice. Van Gasse and Mortelmans (2020a) found that many rejected aspiring parents look for other pathways into single parenthood by choice. Regarding the selectivity in terms of education or income, Harris et al. (2016) showed that the expensiveness of fertility treatments may increase the threshold for lower income groups. This discrimination raises concerns about the socioeconomic equity amongst different social groups. In addition, the age selection of candidates is also associated with their socioeconomic position, because the average age of a first birth is generally lower in groups with a lower socioeconomic status. The screening agents' socially constructed conceptualization of a proper birth age may be influenced by their own frame of reference (Grundy and Foverskov 2016). Research by Van Gasse and Mortelmans (2020b) indicated that a young age is indeed a recurring reason for candidate rejection. Thus, the fertility procedure is not open to everyone, but is structurally focused on a particular group. Despite the existence of open legislation regarding fertility practices, which ostensibly allows anyone below the age of 48 to undertake fertility treatments, the policies seem to miss their purpose (i.e., to be as non-discriminatory as possible) as the desire to become a parent is universal (Hertz 2006).

### 2.5. Doctor Shopping and Unmonitored Fertilization

The gatekeeping dilemma is further complicated by doctor shopping and unmonitored fertilization. First, various studies on single motherhood by choice have suggested that fertility clinics are but one pathway to the single mother family (Van Gasse and Mortelmans 2020b; Hertz 2006; Mannis 1999). Women who can conceive without fertility treatment can opt to forgo the fertility clinic and follow an unmonitored route to planned single motherhood. However, a single mother who follows the unmonitored route does not receive the same preparatory framework as a fertility clinic patient does (Bass 2014). Moreover, those in the unmonitored route did not have to pass any stringent selection criteria. Consequently, the single mothers by choice who might need more monitoring do not receive guidance in the current practice.

Second, a possible side effect of strict gatekeeping procedures without a general legal framework is doctor shopping, as described by Kasteler et al. (1976). The concept of doctor shopping refers to the behavior of patients who consecutively consult different doctors to obtain the treatment they want. Often, this behavior is associated with drug abuse, but

doctor shopping regarding fertility treatments can also be problematic since it undermines the existence of the screening procedure (Martyres et al. 2004). Research by Klitzman (2019) showed how the domain of fertility clinics is vulnerable to doctor shopping. We can argue that the relatively large number of fertility centers in a small space and the transparent legal framework may increase doctor shopping in Belgium. This results in a fertility market rather than decision making based on the principles of parental fitness and the best interest of the child.

Hence, doctor shopping and unmonitored fertilization confront practitioners with an additional dilemma, as both phenomena undermine the decisions of fertility specialists. In the case of negative decisions, women can still become single mothers by choice through other fertility clinics and/or through an unmonitored route. As such, fertility specialists are aware that their decision on parental fitness is not a definitive rejection of parenthood and not final either. We can assume a Matthew effect (a term coined by Merton (1968) to explain accumulated advantage, named after a biblical verse of Matthew (25:29)) in single motherhood by choice as well as a confirmation bias: aspiring single mothers with attributes that score highly in parental fitness have access to the preparatory framework, whereas aspiring mothers lacking these attributes have to find how to construct a single mother family on their own. As a result, single mothers using an unmonitored route may confirm the predefined inequalities in the assessment criteria. All of this further complicates the intra-role conflict.

*2.6. Coping with the Role Strain of Intra-Role Conflict*

We expect the unresolved intra-role conflict to evoke feelings of role strain (Jensen 2016). In their general strain theory, Agnew et al. (2002) described the concept of strain as a form of stress that is derived from the inability to achieve positively valued goals. More specifically, the inability to retain rejected single mother candidates may place fertility practitioners in a disjunction between their own judgement as a gatekeeper and the actual outcomes of the individual fertility-seeking procedure of the aspiring parent (Agnew 1992). Moreover, the role strain is different from stress as it entails a sustaining pattern rather than a temporary outburst. Therefore, strategies also exist to adjust to this remaining feeling of strain.

The way fertility practitioners cope with strain is not known. Therefore, we turned to the literature on general strain coping. In their systematic review, Van den Brande et al. (2016) distinguished two categories of coping strategies and three sources of coping resources. Coping strategies can be generally categorized as problem-focused coping and emotion-focused coping (Van den Brande et al. 2016). In general terms, emotion-focused coping focuses on diminishing the emotional responses to strain, whereas problem-focused coping addresses the cause of the strain (Skomorovsky et al. 2019). Sirgy et al. (2019) argued that an emotion-focused coping strategy can be identified by the presence of venting emotions, denial, and behavioral disengagement, whereas problem-focused coping typically implies planning (i.e., planning how to act when confronted with dilemmas), active coping (i.e., taking active steps to diminish coping), and positive reinterpretations of straining events. Van den Brande et al. (2016) added that wishful thinking and suppression can also be used as emotional coping strategies. Hundera et al. (2020) elaborated the theories of coping with a third coping strategy: the redefinition of the social role. According to Hundera et al. (2020), this redefinition can take place on a personal level (i.e., redefining one's own role), a structural level (i.e., redefining the social or professional role for all related peers), and a reactive level (i.e., adjusting personal expectations to previous outcomes).

The ways in which these coping strategies are used depend on the availability of coping resources. A distinction can be made between personal coping resources, social coping resources, and environmental coping resources (Van den Brande et al. 2016), with the former being an internal resource and the latter two external resources. Personal coping resources are individual characteristics (e.g., personality traits) that help someone cope with strain. Social coping resources are to be found in the construction of one's social network

or culture (e.g., the role of social networks to cope with adverse events). Environmental resources reflect the broader context in which an individual finds him/herself, which helps them to react to strain. As the research of Conner and Bohan (2018) and Skomorovsky et al. (2019) and suggest, the role of peer support, which can be seen as a social resource, proves to be very helpful in a high-strain context. Therefore, the presence of co-worker support can be understood as being a social coping resource. On a more structural level, the possibility of hiring external support via counselling or sharing a workload can be defined as an environmental coping resource (Hundera et al. 2020).

## 3. Methods

We collected ten open in-depth face-to-face interviews with fertility practitioners who were part of the multidisciplinary teams of fertility centers between February 2019 and April 2019. We used a reflexive interview lead (Mauthner and Doucet 2003). This interview lead was derived from a pilot study with 30 single mothers by choice on their experiences and concerns regarding the fertility treatment and from a literature study on the primary concerns described in the literature section. The focus was placed on the various gatekeeping functions of fertility practitioners. Open interview questions were used because they allow for additional answers and for the interview to be adjusted accordingly. The topics of the interview guide included the screening procedure, opinions about the procedure and single parenthood by choice, and their own experiences of the use of the moral clause. The interview duration ranged from 30 min to 45 min. Apart from the interviews, we were able to observe an anonymous staff meeting in one fertility center, to gather valuable data from informal chats kept in memo-documents, and to consult some working documents used in the decision-making process (e.g., a criteria list). The first author was responsible for all the interviews and observations.

Before the interview, every participant received a written explanation of the research purposes and was asked to sign an informed consent. As all the interviewees had a limited time to participate in the interviews, some of them requested to look at the topic list on which the interview lead was based. This topic list included subjects that cover the decision making of the multidisciplinary team regarding single motherhood by choice.

The fertility practitioners who were interviewed worked in one of the 23 fertility centers of Flanders and Brussels, of which 10 participated in this study. An open call for participants was spread across the 23 fertility centers and respondents voluntarily applied. Respondents included any member of the multidisciplinary teams who takes part in the decision-making process and meetings on the subject of single motherhood by choice. Throughout the research process, we iterated the interviews and analysis until a theoretical saturation was reached. In total, six gynecologists, three psychologists, and one midwife were interviewed. All interviews were blinded for colleagues and no details were shared amongst other respondents.

All data sources were first transcribed and centrally collected following the verbatim principles, as described by Van Gasse and Mortelmans (2020a, 2020b), and pseudonymized during transcription. We used NVivo to restructure the interviews and analyze the transcripts, according to the principles of grounded theory analysis (Glaser and Strauss 1967). We used doctor shopping and the unmonitored route as sensitizing concepts and structured the arguments in a first open-coding process towards the different strains in decision making. In this way, we evolved to an overarching concept of intra-role conflict. Next, we used axial coding to explore how individual fertility practitioners deal with the intra-role conflict in the decision-making process, using constant comparison across interviews with individuals performing different roles (i.e., gynecologist, psychologist, and midwife) in the decision-making process and across the different fertility clinics.

## 4. Results

### 4.1. Intra-Role Conflict

In Figure 1, we present our coding scheme of how fertility specialists cope with the described intra-role conflict in fertility gatekeeping. The left part illustrates the intra-role conflict that was expected based on the findings of the existing literature and confirmed by the fertility practitioners. The lean legal framework that grants a high level of agency to the fertility practitioners turns these medical practitioners into selective gatekeepers. Due to the conflicting principles of the best interest of the patient versus the best interest of the child, an intra-role conflict emerges, which is intensified by the risk of patients taking part in doctor shopping or following an unmonitored route.

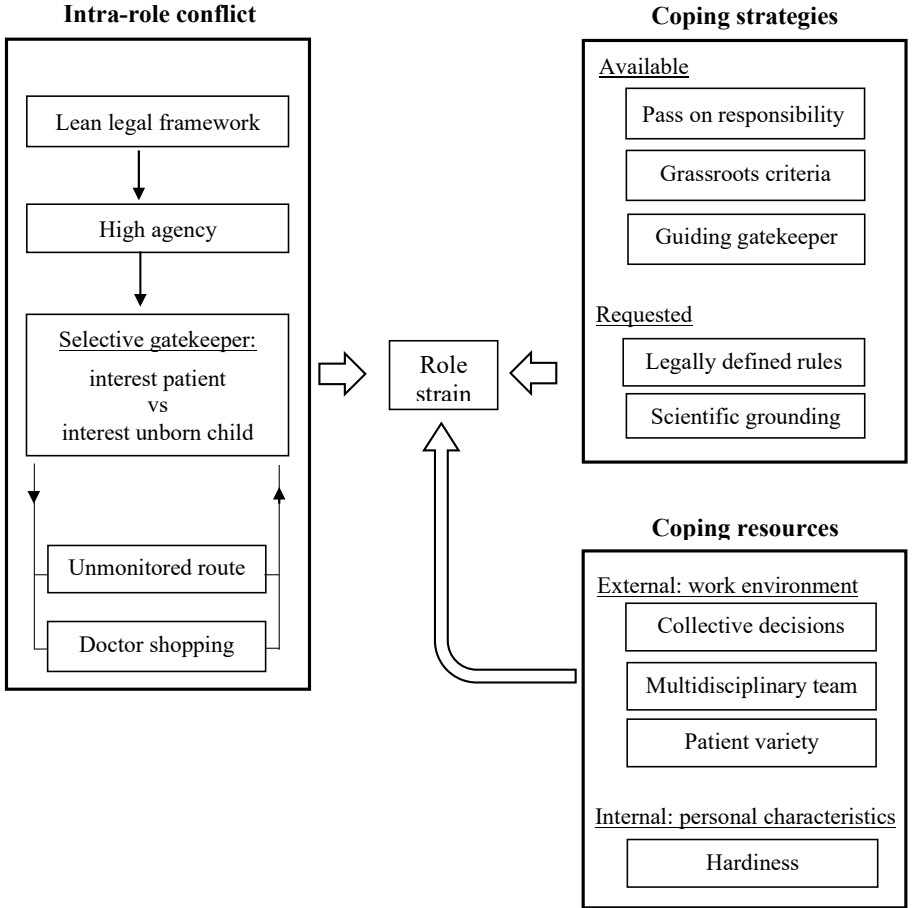

**Figure 1.** The coding scheme of coping with role strain from the intra-role conflict during decision making.

### 4.2. Coping Strategies

We detected three major coping strategies of fertility practitioners: passing on responsibilities, grassroots foundation of selection criteria, and preparing single mothers to meet the defined standards of the practitioner. Fertility practitioners can employ two or all three of these coping strategies simultaneously, but no clear pattern was discovered as to why a practitioner is more inclined to use a specific coping strategy. Regarding the first strategy, there are multiple ways in which fertility practitioners are passing on responsibility. One option is to return the agency to the patient. In these cases, fertility practitioners interpret their role of gatekeeper as being a purely selective assessor. When they are convinced that aspiring mothers are not fit to be a single parent, they reject the patient. However, a gatekeeping role that solely functions as a selective assessor may implicitly nudge clients to go doctor shopping and/or to follow the unmonitored route. We found indications in

interviewees' narratives that they were aware of this implicit nudge, but often assumed that other gatekeepers would take a similar decision when they referred patients to each other. This also implies that there exists an assumption about a shared value system among practitioners on their perspective of *fit single parents.* Therefore, some fertility practitioners chose to pass on the personal responsibility to select a candidate to another fertility practitioner. In addition, they could also pass the responsibility on to social scientific research, in which socio-demographic characteristics related to bad outcomes in children's later life in family studies are interpreted as selection criteria to accept and/or reject candidate parents (e.g., socio-economic status).

> "At one point, we started with inseminations and we got a telephone from another gynecologist saying . . . . "I've sent her to you for an evaluation, but you were meant to reject her!" . . . But we decided as a multidisciplinary team it was okay . . . There is a thin line between referring because you are unsure or referring because you don't want to make a decision." (Gynecologist 4)

Second, practitioners created a grassroots criteria list, with the aim of supporting their decisions regarding acceptance in the fertility programs. This grassroots criteria list was an organically grown assessment of attributes related to parenting competency. The establishment of these criteria was performed by a central person whom the others referred to as "one of the 'anciens'," a Flemish slang referring to an older and more experienced peer (i.e., senior practitioners). This criteria list is regularly updated and partially based on a close reading of research on single-mother families. The use of the grassroots criteria list across fertility centers limits doctor shopping, to some extent, as most fertility clinics assess potential single mothers using the same attributes. Addressing the criteria of parental fitness also does voice a dominant parenthood ideology of what constitutes good motherhood.

> "Well, up till now, there are three eras in selection criteria. First, fertility centers were not prepared for the demand of singles to become parents and no fertility center had developed a way of screening single women with a wish for children . . . Thereafter, we started to develop our own criteria, based on income, social network . . . based on our gut feeling of what is important to manage as a single mother. The last years, an 'ancien' in the field distributes a rudimentary criteria list and I think that this is used in most clinics." (Gynecologist 3)

Third, practitioners argued that they started to shift from a selective gatekeeper perspective towards a guiding stance for single mothers. This reaction was argued to result from the fear that clients would go doctor shopping and behave in accordance with the assessment standards used by the fertility clinics, and the responsibility strain, not willing to push single mothers to the unmonitored route. As a result, some practitioners tended to invite clients to their private practices, or refer them to fellow psychologists to help with "unresolved issues." This way, they prepared aspiring mothers for their role as a single mother and/or referred them to a different fertility specialist to assess the screening once more. The role transformation from a fertility gatekeeper to a fertility guide, helped the fertility practitioners to reconcile the screening and caretaking role that conflict. This reduced the role conflict that fertility practitioners experience. However, it did not reduce the bias towards the dominant parenthood ideologies that fertility practitioners hold, because the criteria towards which they guide aspirant parents are largely the same as the criteria upon which best-fitting candidates are selected.

> "I see certain people with unresolved issues . . . at first, I used to reject those people without further ado but I realized they would try their luck elsewhere, acting to the test and not learning anything . . . so now I invite those clients to my private practice to deal with the unresolved issues and at the end of trajectory, I ask whether they still want to become a single mother by choice." (Psychologist 3)

Besides these available coping strategies, various interviewees expressed the need for a framework supporting the decision-making process of fertility practitioners. This

supportive framework would rely both on a legal-aligned framework and on a scientific foundation based on studies on the long-term consequences of single motherhood by choice. Regarding legislation, the views of fertility specialists were surprisingly diverse. Whereas some addressed the need for protective measures towards claims of donor-conceived children on fertility practitioners' responsibility regarding their birth, others took a negative stance regarding policy interference. Nevertheless, all of our interviewees referred to the need for supplementary scientific research supporting screening and assessment criteria of aspiring single mothers.

> "We don't need an additive legislation as such, but we need scientific grounding. We have to address the question 'What is a good parent?,' but yes, what does make a parent a good parent?" ). (Psychologist 2)

Interestingly, respondents both wish more grounding for the objectivity in their screening criteria and assure that the interviewer that the assessment is objective and based on a shared screening practice. This shows that the screening guidelines are emergent and there efficacy might need to be evaluated. There appears to be much insecurity about the assessment criteria in the current practices.

### 4.3. Coping Resources

Next to the coping strategies described earlier, there are also external and internal coping resources that help protect against some of the role strain fertility practitioners face. External coping resources consist of certain organizational attributes of their work that prevent or alleviate role strain, whereas internal coping resources reflect more personal attributes that form a protective barrier.

Regarding the external resources, interviewees referred to the possibility of taking decisions collectively, being a multidisciplinary team, and the variety of patients (i.e., not only single women, but also couples and patients of different socio-economic and cultural backgrounds) at the fertility clinic. First, psychologists, doctors, and midwives often reflected on the fact that they did not make the choice on their own. The decision to reject or accept aspiring single mothers is made as a team, based on multiple concerns shared by different agents. Secondly, these teams are multidisciplinary in nature, meaning that team members look at the decision from different vantage points, (e.g., gynecologist, psychologist, etc.). This collective decision making by different professions helped gatekeepers in their self-awareness of having taken a decision objectively.

> "I don't make the decision, nor does the doctor. We discuss every case as a team, with all the doctors, midwives, and psychologists. As a psychologist, I address my concerns and these are discussed. Sometimes, positive cases from the past erase my doubts, but we usually find a decision that we feel is a good one." (Psychologist 1)

A third external resource consists of the variety in patients. The job of a fertility practitioner does not solely consist of rejecting or accepting aspirant single mothers. A great deal of their work revolves around guiding couples into parenthood. Couples do not face the same stringent admission tests. This seems to be a relief for some respondents, not having to constantly judge patients, but instead to guide them. It also means that the tension of deciding on single parenthood by choice is limited and not the only task of the fertility practitioners.

> "You know we are talking about single motherhood by choice now, and that is always a very difficult case. But there are other things as well, we help infertile couples to live their dream as parents, or help same-sex couples to become a parent. A single mother case is always something different, but if we have doubts, we can always talk to each other in this center." (Midwife 1)

Next to these environmental attributes, there are internal resources, helping gatekeepers to cope with the role strain. A common trait in the gatekeepers who had to make the

final decision was hardiness, taking the plunge when necessary. For some, hardiness at work was only possible if relief could be found at home.

> "I can get very rough in here, against patients and colleagues ... I am the one stating "over my dead body" when a colleague defends a potential candidate that I am seeing totally unfit. But, we are taught to respond to "help me," and saying no is against the nature of any doctor so at home, I vent and I try to do fun stuff." (Gynecologist 6)

When confronted with those hard decisions, fertility practitioners elaborate on how rejecting potential candidates defy their original practice of helping patients. Instead, they find support in their home environment. A space to vent and can be assumed to be relevant to maintain a proper work–life balance and process the events of the day, in which they were confronted with these role conflicts.

## 5. Discussion

The findings of this study show that fertility practitioners in Belgium mitigate role strain with the help of coping strategies and the availability of coping resources. Regarding coping strategies, Van den Brande et al. (2016) distinguish problem-focused and emotion-focused coping strategies. We find that fertility practitioners use both problem-focused coping and emotion-focused coping (Skomorovsky et al. 2019). Problem-focused coping is most visible in the actions used in the work routines at the fertility clinic. Within this area, we found that fertility practitioners had three main strategies available to cope with strain: passing on responsibilities, building grassroots criteria lists, and shifting their role from a gatekeeper to a guide. Passing on responsibilities and a grassroots collective criteria list were used in all fertility centers to cope with the intra-role conflict confronting practitioners, and some fertility practitioners decided to change their role. Aside from these available coping strategies, the fertility practitioners requested more scientific grounding or, to a lesser extent, a legal framework to ease their role strain as possible coping resources. The different coping strategies can be used individually or in combination.

First, the act of passing on responsibilities entails, from a positive perspective, support-seeking behavior (Skomorovsky et al. 2019). When fertility practitioners pass on responsibilities, they look for empowerment for their decision making and, as Conner and Bohan (2018) stated, interacting with peers and mentors can help to build confidence in decision making and cope with strain in everyday life. In a more negative form, this act can also result in avoidant coping. By passing on the responsibility to a fellow practitioner, gatekeepers avoid fulfilling their screening function. This can lead to role retreatism: actively backing away from the gatekeeping role (Murray 1983). In an ideal scenario, passing on responsibilities helps fertility practitioners to make decisions when they are insecure. Because of the absence of shared guidelines, passing on responsibilities might lead to doctor shopping, which only further aggravates the intra-role strain.

Second, the grassroots criteria list is an example of an active coping strategy, which Sirgy et al. (2019) defined as "planning behavior". A grassroots criteria list, i.e., the shared guidelines, can result in homogeneous procedures throughout different fertility centers, reducing the risk of doctor shopping. With the criteria list, the strain related to future decisions is reduced, as the decision making is based on mutually constructed criteria of what the gatekeepers deem to be indicators of parental fitness. Nonetheless, the conception of the different criteria can be problematized as well since it is based on the shared normative expectations and parenthood ideologies of the fertility practitioners. To construct this criteria list, the gatekeepers also look for support from each other and learn from mentors, aspects Conner and Bohan (2018) and Skomorovsky et al. (2019) regarded as important behaviors to reduce strain. It also offers a way to reaffirm past decisions, when in doubt, which is one of the problem-focused coping strategies Van den Brande et al. (2016) described.

Third, our finding that fertility practitioners redefine their roles resonates with Hundera et al. (2020), who suggested role redefinition as a possible coping strategy. Re-

spondents argued that they would rather guide aspirant parents to the criteria of parental fitness than selecting the best-fitting candidates, because the latter would nudge rejected candidates towards doctor shopping and the unmonitored route. However, this role re-definition requires more effort from the gatekeepers as they must combine multiple roles, which can lead, in its turn, to inter-role conflict. Hence, a structural role redefinition would imply a structural increase in means.

*5.1. Implications for Parenting in the 21st Century*

Through increasing possibilities in ART, fertility practitioners bear an important role voicing parenthood ideologies in the 21st century (Gürtin and Faircloth 2018). For fertility practitioners, norms and values about parenting are not only performative in day-to-day behavior (e.g., Van Gasse et al. 2021), but also selective in the screening procedures they hold. Current policies and medical guidelines still lag behind day-to-day practices. Attention has to be drawn to these practices since possibilities to create families may have been moved beyond the nuclear ideal, but the question of how to assess the fitness of aspiring parents to head those families remains unanswered (Cutas and Chan 2012). Our study shows that bearing the responsibility to assess, screen, and select aspiring parents based on parental fitness criteria can result in intra-role strain for which different coping strategies exist.

Current screening practices may also be criticized given the inherent gatekeeping nature that is discussed in this study (Storrow 2006). The laisser-faire standpoint that is embedded in Belgian policies allows dominant parenthood ideologies to shape the demographics of single parents by choice (Sperling and Simon 2010). Whereas the assessment of practitioners is deemed to be objective, it remains unclear to what extent the screening criteria are based on objective assumptions (Johnson 2012). This may explain the typical socio-demographic profile of women who became a parent through ART (Graham 2012).

The screening criteria remain an issue for fertility practitioners as well as for outsiders. Whereas gatekeepers look for scientific grounding to positively reinterpret their reasoning behind the gatekeeping criteria, outsiders might experience the current practice as arbitrary, which was also found in the studies of Van Gasse and Mortelmans (2020b) and Peterson (2005).

*5.2. Policy Implications*

The findings of this study are not only of interest to the social scientific community, but can also offer insight to policy makers. First, there exists a large amount of knowledge on the life of single-parent families that can be shared with fertility practitioners. In Flanders, the life of single-parent families has been widely documented over the last decades, although relevant studies have focused predominantly on single parenthood by divorce (Mortelmans et al. 2011). Research in Flanders, as well as in an international context, has suggested that single-parent families can be functional family units (Van Gasse and Mortelmans 2020a; Morrison 1995). However, the findings of this family research might not be known by fertility practitioners and thus needs to be communicated more effectively. This can be enabled by investing in open access publications from the family research departments and developing interdisciplinary research networks between fertility researchers and family researchers to jointly distribute relevant knowledge to the specific case of single parenthood by choice. There seems to be a foundation to approach single parenthood by choice not only as a medical, but also as a psychological and social issue. A second recommendation addresses policymakers with the need for a more elaborate legal framework regarding single parenthood by choice. As single parenthood by choice was not addressed in the legal fertility framework, policymakers in effect created a laisser-faire system (Schiffino and Varone 2004). As a reaction and coping strategy towards the role strain, fertility practitioners started a grassroots collaboration to construct gatekeeping criteria. These gatekeeping criteria can work very well in a context in which there is a good communication between different fertility centers, preventing doctor shopping. However,

at the moment, the gatekeeping criteria are only practical guidelines and not legally binding. Therefore, we recommend Belgian policymakers to implement adjusted legislation based on the informal gatekeeping criteria in a close collaboration with fertility practitioners and family researchers. A transparent and homogeneous legislative framework may support fertility practitioners in a transition from selective assessor to preparative gatekeeper, helping their patients and also decrease the tension between patients and practitioners since they are less perceived in a judging role, but more as a caretaker (Van Gasse and Mortelmans 2020b). Although additional funding of fertility centers might be needed to finance the extra tasks that a preparative gatekeeper needs to perform, these policies may be able to tackle the current selectivity taking place in fertility clinics.

Aside from the current study, a few existing studies (i.e., Okonta et al. (2018); Merchant (2019), and Pawa et al. (2020)) have demonstrated that similar issues also reside in other countries apart from Belgium. Although transnational guidelines might offer a solution, they can be inefficient when countries differ a lot, e.g., Belgium and Nigeria (see Okonta et al. (2018)). Consequently, we support international guidelines when they apply to nations with similar contexts. The European Society of Human Reproduction and Fertility offers such guidelines for several topics, but currently not for the assessment of single parents by choice via ART (Gameiro et al. 2015).

*5.3. Limitations*

Our study has four major limitations. First, our study was limited by the small research population. Hence, the witness accounts of fertility practitioners all have a very strong impact on our results. Second, due to the compartmentalized and trilingual nature of the Belgian country, we chose to focus on fertility centers in the Flemish and Brussels region. Therefore, possible regional disparities between the Flemish and Walloon fertility centers were not included in our analyses. Third, different interviewees referred to the original writer of the grassroots criteria list, but we did not have access to this important actor in the field of Belgian fertility clinics. Therefore, we also had no first-hand account of the construction of this criteria list. Fourth, the disproportionally high representation of gynecologists in our research population may, to some extent, bias our results. Although these were often the main decision makers on the multidisciplinary teams, they may look differently at the gatekeeping process than, for example, midwives, of whom only one agreed to participate as an interviewee.

*5.4. Future Research*

This research is one of the first studies on single mothers by choice focusing on the perspective of the practitioner. However, as it answered our main research question (i.e., how do fertility practitioners cope with the strain resulting from intra-role conflict in the decision-making process of single motherhood by choice in Belgium?), it opened up many questions that future researchers should address. First, future research should examine how both single mothers impregnated in a fertility clinic and single mothers following an unmonitored route differ, and which mothers successfully adjust to a single-parent-family life. As such, it should be documented on which attributes can be focused in preparation programs. Second, there is a gap in the literature in the formulation of rudimentary parenting attributes. Future research should address what the minimal needs are to address parental fitness in screening procedures. As it is impossible to close unmonitored routes to single motherhood by choice, research and practitioners should focus on preparing single women for their parenting challenge, rather than on selecting the best candidates and nudging others towards a different pathway.

**Author Contributions:** Conceptualization, D.V.G. and S.S.; formal analysis, P.H., D.V.G. and S.S. investigation, P.H., D.V.G. and S.S.; resources, D.M., data curation, D.V.G.; writing—original draft preparation, D.V.G. & S.S.; writing—review and editing, D.V.G., S.S. & D.M.; visualization, S.S.; supervision, D.V.G. & D.M.; funding acquisition, D.M. All authors have read and agreed to the published version of the manuscript.

**Funding:** This research was funded by FONDS WETENSCHAPPELIJK ONDERZOEK, grant number 140069 and grant number S002719N.

**Informed Consent Statement:** Informed consent was obtained from all subjects involved in the study.

**Data Availability Statement:** Data is unavailable for external readers due to privacy reasons.

**Conflicts of Interest:** The authors declare no conflict of interest.

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
