# Peer review of "Fertility Practitioners’ Coping Strategies When Faced with Intra-Role Conflict from Screening Aspiring Single Mothers by Choice"

_socsci, doi:10.3390/socsci10110438_

Round 1

Reviewer 1 Report

This study aims to explore the coping strategies enacted by fertility experts to deal with intra-role conflict when deciding to accept or to reject the provision of treatments to single mothers by choice. This is an interesting topic with potential to add valuable insights to rethink legal frameworks and guidelines.

There are however multiple issues with the manuscript that should be addressed before the paper can be considered. I have two major concerns. First, readers might expect a more conceptual 'think-piece' that critiques the practices of fertility experts and the underlying sociocultural normative expectations and morality about the identities and roles played by women in relation to parenting. The study as presented is very descriptive in nature and lacks analytical strength. Second, the presentation and discussion of the results are more like a laundry list than true “findings”. More work could be done to locate the findings inside most of the debates on the topic across the social sciences.

Specific comments

1. Lines 34-36: There are single mothers by choice who do not use “clinical fertilization”.

2. Clinical practitioners might invoke moral clauses in several circumstances. It is mostly used to reject specific procedures and not to reject particular patients/candidates. It would be helpful to explore this tension throughout the manuscript (attitudes towards single motherhood by choice vs. individual assessment of a specific candidate).

3. The section “fertility legislation in Belgium” could be synthesized by focusing on the data relevant to the topic of the paper.

4. Line 109: Please clearly identify the “competing interests” reported in the literature. Afterwards, findings could explore the similarities and differences between them and the competing interests mentioned by the interviewees. What is the role of experts’ values/professional ethics towards some “structural” factors (e.g. doctor shopping)?

5. Lines 112 ss.: Does this “very specific profile” of “accepted patients” result from the selection made by fertility experts? Or does it correspond to the profile of women who search for fertility treatments?

6. Line 169: What is meant by “fair-just outcomes and actual outcomes”?

7. Please include the following information on the methods section: period of data collection; face-to-face/digital/other type of interviews; number of interviewers; who was invited – fertility centers or specific fertility experts; topics of the interview guide; data saturation; ethics approval. How did you deal with anonymity and privacy of the interviewees (in particular when there is only one midwife)? Was there enough new and in-depth data to allow for a grounded theory analysis?

8. Line 241: Is “bioethical dilemma of fertility gatekeeping” synonymous of “intra-role conflict”?

9. Figure 1 seems to be more a coding scheme than a conceptual model.

10. Lines 251 ss.: Do the three major coping strategies occur simultaneously? Its enactment depends on what – fertility clinic, interviewee, candidate?

11. Line 282: Please identify all the “attributes” used in the grassroots criteria list.

12. Why do you classify “the variety in patients” as an external resource? This “variety” seems to be grounded on a single characteristic - marital status (single versus couple). Some respondents considered this as “a relief”, but it can also be the way around (feelings of pressure, stigmatization, unfair/disproportionate assessment, etc.).

13. More sense-making work needs to be done to discuss findings and highlight key points that really address a gap in the literature.

14. Lines 457-459: Social science studies “might not be known by fertility practitioners”, thus authors recommend a “more effective communication”. How to communicate “more effectively”? Who is responsible for that – social scientists, fertility practitioners, or both?

15. Lines 465-466: Why do you recommend an “adjusted legislation” and not the provision of transnational guidelines? 

Author Response

Reviewer 1

Dear reviewer,

Thank you for your assessment of our manuscript entitled: ” Fertility gatekeepers’ Coping Strategies when Faced with Intra-role Conflict from Screening Single Mothers by Choice”. We appreciate your interest in our research topic and overall appreciation of the manuscript. We do, however, also understand your concerns. In the next alineas, we will discuss how we altered our manuscript based on your concerns in the hope to meet the expectations you have about the paper. We will cite your comments in bold text and address our adaptions in a point-wise manner.

First, readers might expect a more conceptual 'think-piece' that critiques the practices of fertility experts and the underlying sociocultural normative expectations and morality about the identities and roles played by women in relation to parenting. “

We appreciate this comment, we revised the paper thoroughly to make this possible. As an example, we added a new Alinea “Single mothers by choice challenging the conception of parenthood in the 21st century”in which we problematised the screening procedures more in a theoretical way.

We also felt that we have the material and perspective to write this article as an explorative empirical paper and not as a conceptual critique. In order to emphasize this more clearly, we mention the explorative nature of our study in the abstract of this revision.

“The study as presented is very descriptive in nature and lacks analytical strength. Second, the presentation and discussion of the results are more like a laundry list than true “findings”. More work could be done to locate the findings inside most of the debates on the topic across the social sciences.”

We understand that this study feels descriptive and might resemble a list of results. However, given that this study examines a previously unexplored subject (i.e., Belgian fertility practitioners’ role conflict due to their gatekeeping role when screening aspirant single mothers), we believe that the descriptive nature of this explorative study is not unjustified. In addition, we have extensively revised the discussion section, which now transcends descriptive results and offers a conceptual critique of the discrimination that is inherent to the Belgian case.

  1. Lines 34-36: There are single mothers by choice who do not use “clinical fertilization”.

This comment is, off course, true. We added a sentence to clarify why clinical fertilization is different than other options such as adoption: “The option of clinical fertilization stands out because it allows single women to conceive (in contrast with adoption) without the physical presence of a male donor.”

  1. “Clinical practitioners might invoke moral clauses in several circumstances. It is mostly used to reject specific procedures and not to reject particular patients/candidates. It would be helpful to explore this tension throughout the manuscript (attitudes towards single motherhood by choice vs. individual assessment of a specific candidate).”

Whereas we think this can be a valuable addition to the paper, we don’t have the data to elaborate on this. All our participants worked at fertility clinics that specifically facilitated single parenthood by choice. We did, however, referred to the study of Okonta et al. (2018) who did delve deeper into this topic and related our findings to theirs. We discuss this study later in this letter as well.

  1. “The section “fertility legislation in Belgium” could be synthesized by focusing on the data relevant to the topic of the paper.”

Here, we had to make decisions in the contrasting comments of both reviewers since reviewer 2 argued more context about Belgium was needed. We opted to situate Belgium from a cultural perspective, which was suggested by reviewer 2 and synthesised the legislation to what we felt was the essence (removing the policy making context). We hope this suffices but will gladly make other additions if needed.

  1. Line 109: Please clearly identify the “competing interests” reported in the literature. Afterwards, findings could explore the similarities and differences between them and the competing interests mentioned by the interviewees. What is the role of experts’ values/professional ethics towards some “structural” factors (e.g. doctor shopping)?

We acknowledge that such a comparison is highly interesting. However, since our study is the first to examine intra-role conflict among fertility practitioners working with single mothers by choice, we cannot compare the competing interests found in our findings to those of the existing literature. 

  1. Lines 112 ss.: Does this “very specific profile” of “accepted patients” result from the selection made by fertility experts? Or does it correspond to the profile of women who search for fertility treatments?

We understand that this argument seems to resemble a ‘chicken or egg’ argument, but we base the selectivity argument on the work of Rosanna Hertz (2006) who claims that single mothers by choice are a diverse group who rather find themselves single by ‘chance’ (and choose specifically for motherhood) and the idea of different pathways to single parenthood by choice that were explored in Van Gasse & Mortelmans (2020). Moreover, it is documented that ART is expensive and therefore less attainable for women with a lower income, which also scares off lower ses women. We hope the following new lines clarify that it is the selection made by fertility practioners and the cost of the fertility treatments that result in a very specific profile of accepted patients.: “According to Hertz (2006), this profile does not match the general public of candidate single parents and Van Gasse & Mortelmans (2020) found that many rejected aspiring parents reported to look for other pathways to single parenthood by choice. Also regarding the selectivity in terms of education or income, Harris et al. (2016) showed that the expensiveness of fertility treatments may increase the threshold for lower income groups. This discrimination raises concerns about the socioeconomic equity amongst different social groups”

  1. Line 169: What is meant by “fair-just outcomes and actual outcomes”?

We agree that this phrasing was unclear from our side. Therefore, we rephrased the sentence as follows: “More specifically, the inability to retain rejected single mother candidates may place fertility practitioners in a disjunction between their own judgement as a gatekeeper and actual outcomes of the individual fertility seeking procedure of the aspiring parent (Agnew, 1992).”

  1. “Please include the following information on the methods section: period of data collection [added]; face-to-face/digital/other type of interviews [added]; number of interviewers [added]; who was invited – fertility centers or specific fertility experts [added]; topics of the interview guide [added]; data saturation [added]; ethics approval. How did you deal with anonymity and privacy of the interviewees (in particular when there is only one midwife) [added]? “

We added all these details about our methodological procedure in the method section. The details can now be found in the following Alinea: “We collected ten open in-depth face to face interviews with Fertility practitioners taking part in the multidisciplinary teams of fertility centers in between February 2019 and April 2019. We used a reflexive interview lead (Mauthner and Doucet, 2003). This interview lead was derived from a pilot study with 30 single mothers by choice on their experiences and concerns regarding the fertility treatment and a literature study on the primary concerns described in the literature section. The focus was placed on the various gatekeeping functions of fertility practitioners. Open interview questions were used because they allow for additional answers and for the interview to be adjusted accordingly. The topics of the interview guide included the screening procedure, opinions about the procedure and single parenthood by choice and own experiences of usage of the moral clause. The interview duration ranged from 30 minutes to 45 minutes. Apart from the interviews, we were able to observe an anonymous staff meeting in one fertility centre, to gather valuable data from informal chats kept in memo-documents, and to consult some working documents used in the decision making process (e.g., a criteria list). The first author was responsible for all the interviews and observations. Before the interview, every participant received a written explanation of the research purposes and was asked to sign an informed consent. As the interviewees had limited time to participate in the interviews, some of them requested to look at the topic list on which the interview lead was based. This topic list included subjects that cover the decision making of the multidisciplinary team regarding single motherhood by choice. (…) All data sources were first transcribed and centrally collected following the verbatim principles, as described by Mortelmans (2020) and pseudonymized during transcription. We used NVivo to restructure the interviews and analyse the transcripts, according to the principles of grounded theory analysis “

  1. “Line 241: Is “bioethical dilemma of fertility gatekeeping” synonymous of “intra-role conflict”?”

These are indeed synonyms. We changed this throughout the manuscript.

  1. “Figure 1 seems to be more a coding scheme than a conceptual model.”

We adjusted this and now use “coding scheme” instead of conceptual model.

  1. Lines 251 ss.: Do the three major coping strategies occur simultaneously? Its enactment depends on what – fertility clinic, interviewee, candidate?”

This is again a very valid question. We have added our answer to the revised text: “Fertility practitioners can employ two or all three of these coping strategies simultaneously,but no clear pattern was discovered as to why a practitioner is more inclined to use a specific coping strategy.”

  1. “Line 282: Please identify all the “attributes” used in the grassroots criteria list.”

Whereas we agree that it is interesting to look deeper into the attributes used in the grassroots criteria list, this list was not accessible for us as researchers due to the confidential nature of the assessments the fertility experts made. Therefore, we cannot elaborate on the attributes used.

  1. “Why do you classify “the variety in patients” as an external resource? This “variety” seems to be grounded on a single characteristic - marital status (single versus couple). Some respondents considered this as “a relief”, but it can also be the way around (feelings of pressure, stigmatization, unfair/disproportionate assessment, etc.).”

We understand that this was vaguely phrased. We added a sentence in between brackets to make it more clear what we mean with a variety in patients: “(i.e. not only single ; women but also couples, and patients of different socio-economic and cultural backgrounds)”. We also added a sentence to explain why this was perceived as an external resource to cope with the role strain: “It also means that the tension of deciding on single parenthood by choice is limited and not the only task of our respondents.”

  1. More sense-making work needs to be done to discuss findings and highlight key points that really address a gap in the literature.

Thank you for this comment, we reworked the discussion and theory section to reconnect them in a better way. We hope these changes work meet the requirements you have about the paper.  

  1. “Lines 457-459: Social science studies “might not be known by fertility practitioners”, thus authors recommend a “more effective communication”. How to communicate “more effectively”? Who is responsible for that – social scientists, fertility practitioners, or both?”

This point is well taken. In our opinion, the responsibility lies not alone with social scientists nor with fertility practitioners. We believe that there is fertile ground to establish interdisciplinary research on the topic of single parenthood by choice to approach this not only as a medical, but also as a psychosocial matter. Therefore, we added: “This can be enabled by investing in open access publications from the family research department and developing interdisciplinary research networks between fertility researchers and family researchers to jointly distribute relevant knowledge to the specific case of single parenthood by choice. There seems to be fertile ground to approach single parenthood by choice not only as a medical, but also as a psychological and social issue”

  1. Lines 465-466: Why do you recommend an “adjusted legislation” and not the provision of transnational guidelines? 

In reworking the manuscript, we stumbled upon  articles stating similar findings as we described. We now understand that transnational guidelines might be needed indeed. Therefore, we added the sentence:  “we strongly recommend an adjusted legislation based on the informal gatekeeping criteria, in a close collaboration with fertility practitioners and family researchers or even transnational guidelines given that other studies (e.g. Okonta et al. (2018), Merchant (2019) and Pawa et al. (2020)) address similar issues in other national contexts”

In reworking the manuscript, we stumbled upon  articles stating similar findings as we described. We now understand that transnational guidelines might be needed indeed. Therefore, we added the sentence:  “Aside from to current study, a few existing studies (i.e.. Okonta et al. (2018), Merchant (2019) and Pawa et al. (2020)) have demonstrated that similar issues also reside in other countries beside Belgium. Although transnational guidelines might offer a solution, they can be inefficient for widely varying countries, e.g. Belgium and Nigeria (see Okonta et al. (2018)). Consequently, we support international guidelines among nations with similar contexts. The European Society of Human Reproduction and Fertility offers such guidelines for several topics, but currently not for the assessment of single parents by choice via ART (Gameiro et al., 2015).”

Reviewer 2 Report

The article is original in its approach and research focus, and with major revisions will be a good contribution to the interdisciplinary and multi-disciplinary research field of assisted reproduction, and other relevant fields. It is my sincere hope that the author(s) will revise and also partly restructure the text to strengthen its main point, and to clarify findings. Below, I raise some issues that I view as of major concern, and that I think need to be addressed before this article is ready for publication.

The use of the term “experts” in the article is not explained. Usage glides between experts and specialists, and practitioners. The preference for “experts” (and the fact that it appears in the title) suggests an underlying question about expertise: while the practitioners in question are undoubtedly experts in their fields (reproductive health) – to what extent are they also experts in assessing prospective single mothers? Current praxis places them as such “assessment experts,” for sure. This term could be explained and to some extent problematized in the article.

The study is based on a small sample of only ten interviews, and with a mix of professions. Whereas the inclusion of several professions is fine in itself, it is unclear from the methods section what motivates this sample. Of a possible 120 persons active in Belgium (according to the article), only 10 have been interviewed. Why? This is not explained, nor is the number of interviews motivated in the text. This should be done, and the claim on lines 476-477 “we were able to include large part of the population” does not seem legitimate, given the size of the sample.

The cultural/national context of Belgium is highly interesting, both for its near-lack of legislation on the matter at hand, which is discussed in the article – but also for its particular cultural make-up – which is not discussed in the article -- as a multilingual and multiethnic country, with large groups of both protestants and catholics (usually an important factor that impacts issues and values around reproduction, parenthood and so on). Perhaps authors signal more than language differences when they refer to the areas in which professionals work – but this may be less obvious to the international reader? For an international journal such as Social Sciences, context of Belgium specifically should be a bit more fleshed out to allow international readers to understand more about reproductive practices and potential conflicts of interest.

Very interesting data is presented. At times it is not discussed or analysed however, but “left on its own” – for example when a psychologist is quoted as raising the question about meanings of “good mother”, and when the gynecologist presents themselves as the one who “puts their foot down”. Also, the data as presented now, in quotations from interviews, suggests certain ideas are voiced by practitioners which are NOT discussed in the article, but could be! For example:

  • Assumptions about shared values (the example when one practitioner refers a woman to another practitioner, passing on the responsibility, and expecting that the other practitioner will share the same values and make the same assessment. This is linked of course to the “criteria list.” [the “original writer” of this list is mentioned on line 481, but this is a bit cryptic?]
  • Assumptions regarding meanings of “good mothers” (linked to risks for children, and possibly also mothers). Concerns regarding who should be allowed to mother as a single woman surface here and there, the discussion could be brought together more clearly, also possibly linked to the criteria list. NB there are some unclear passages on this item: for example lines 137-140: what is it the author(s) wants to convey here? The reasoning is not transparent, nor is the meaning of mothers “needing the most monitoring”? Are authors suggesting that mothers who are rejected e g due to lacking financial resources will not manage motherhood without “monitoring” by experts? If so, what is this assumption based upon?
  • Ideas and concerns about risk for oneself as a professional, from making certain assessments and recommendations.

Also, the assumption by (some) of the experts that they make “objective” assessments is highly interesting, and I suggest the author(s) critique/problematize this. See e g line 336, and compare to the quote just before, on lines 319-321. Could be interesting to bring this together with discussion of gatekeeper role and retreating from gatekeeper role on page 9.

So far, I have mentioned 1) clarification of a central concept (experts); 2) clarification of rationale behind sample, and adjusting claims to sample size; 3) strengthening contextualization by building it up slightly more, with an international readership in mind; 4) making sure the quotations/data/results are more thoroughly commented on and critically addressed in the analysis.

I would also add that the authors’ strong recommendation for “adjusted legislation” (page 10) based on the criteria list is a bit surprising, given that the interviewees do not raise further legislation as a felt need. Perhaps the argument for this could be built up to become more persuasive? Beyond these items, I have only a few comments on presentation/structure, and language.

There is quite some repetition in the article, and I suggest some deletions of repetitive passages and sections, in order to make room for developing the main argument more fully.

  • For example, the 1st in Results repeats things that are presented in Background; perhaps they should NOT be in Background.
  • Also, under Discussion – the 1st is repetitive, it is only in §2 (“The findings of this study” lines 384 ff) that we get to actual discussion, start here!

The article is generally well written and has fluency, but language editing will be necessary. Below are some examples of the areas that are problematic: sentence structure, word choices, grammar, prepositions.

Sentence structure: Lines 131-133: messy sentence, please revise.

Word choices, e g Line 10 (abstract) result? Might mean “resort”?; Line 47 educate a child? Is this what parents are expected to do?

Grammar/syntax: Line 125 the candidates’ rejection – sounds as if it is prospective mothers who are rejecting; please rephrase to convey that it is the experts who reject candidates.

Prepositions: Line 7 (abstract) in account should be into account; Line 394 Aside for should be aside from.

All best, your reviewer.

Author Response

Reviewer 2

Dear reviewer

Thank you for the review of our article and the general appreciation. We also value your comments highly and think that addressing each and every issue helps us to reach a better quality in our writing. We will discuss each comment point-wise and address how we attempted to tackle the raised concerns.

  1. “The use of the term “experts” in the article is not explained. Usage glides between experts and specialists, and practitioners. The preference for “experts” (and the fact that it appears in the title) suggests an underlying question about expertise: while the practitioners in question are undoubtedly experts in their fields (reproductive health) – to what extent are they also experts in assessing prospective single mothers? Current praxis places them as such “assessment experts,” for sure. This term could be explained and to some extent problematized in the article.”

This point is very well taken. We were not yet aware of the linguistic bias the usage of ‘fertility experts’ could provoke. This comment made us contemplate about a more sensible choice of wording and now we use “fertility practioners”.  We introduce the term now as follows: “The decision to invoke a moral clause is motivated by a multidisciplinary team, which members we will refer to as fertility practitioners. This term is preferred rather than fertility doctor or fertility specialist, because the multidisciplinary team also includes members without a medical background, such as psychologists..” We also ensured that we use the term consistently across the manuscript.

  1. “The study is based on a small sample of only ten interviews, and with a mix of professions. Whereas the inclusion of several professions is fine in itself, it is unclear from the methods section what motivates this sample. Of a possible 120 persons active in Belgium (according to the article), only 10 have been interviewed. Why? This is not explained, nor is the number of interviews motivated in the text. This should be done, and the claim on lines 476-477 “we were able to include large part of the population” does not seem legitimate, given the size of the sample.”

This comment is well taken. In the previous version of the manuscript, we forgot to mention that we iterated new interviews until we found theoretical saturation in our Grounded Theory model. We have added this information and removed the part about extrapolation since we felt this was no criterium to evaluate the quality of our analysis.

  1. “The cultural/national context of Belgium is highly interesting, both for its near-lack of legislation on the matter at hand, which is discussed in the article – but also for its particular cultural make-up – which is not discussed in the article -- as a multilingual and multiethnic country, with large groups of both protestants and catholics (usually an important factor that impacts issues and values around reproduction, parenthood and so on). Perhaps authors signal more than language differences when they refer to the areas in which professionals work – but this may be less obvious to the international reader? For an international journal such as Social Sciences, context of Belgium specifically should be a bit more fleshed out to allow international readers to understand more about reproductive practices and potential conflicts of interest.”

We understand that a better contextualisation might be fitting. Therefore, we added this Alinea in the paragraph on the Belgian country context: “The study took place in Belgium, a country we will first situate and contextualize from a cultural perspective, since this can have an impact on the dominant parenthood ideology. Thereafter, we describe the practice in fertility clinics. From a religious perspective – a standpoint which can have an impact on the norms and values on parenthood, reproduction and so on – Belgium has a catholic tradition, which shows in 57,8% of the population that still affiliates with the catholic church (Sealy & Modood, 2019). Although immigration increase since the 1970s has spurred cultural diversity, the second and third biggest affiliations in Belgium are agnosts (20,2%) and atheists (9,1%). The catholic tradition is also partially embedded in Belgian politics with a Christian party that had a vast prominence in the political landscape. Nonetheless, the strong catholic tradition of this party has shifted in recent decades towards a more humanist identification and discourse (Koutroubas et al., 2009). Due to its multicultural and multilinguistic diversity, Belgium has a complex socio-politival structure with three territiorial communities: Brussels, Wallonia and Flanders and three linguistic communities: the French, Dutch and German (Pew, 2017).”

  • “Assumptions about shared values (the example when one practitioner refers a woman to another practitioner, passing on the responsibility, and expecting that the other practitioner will share the same values and make the same assessment. This is linked of course to the “criteria list.” [the “original writer” of this list is mentioned on line 481, but this is a bit cryptic

We adjusted and rephrased this bit, in a couple of sentences in which we address the idea of shared values. This can now be read in the lines between 331 – 334: “We found indications in interviewees’ narratives that they were aware of this implicit nudge, but often assumed that other gatekeepers would take a similar decision when they referred patients to each other. This also implies that there exist a shared value system in between practitioners on their perspective of fit single parents. Therefore, some individual chose to pass on the personal responsibility to select a candidate.”

  • “Assumptions regarding meanings of “good mothers” (linked to risks for children, and possibly also mothers). Concerns regarding who should be allowed to mother as a single woman surface here and there, the discussion could be brought together more clearly, also possibly linked to the criteria list. NB there are some unclear passages on this item: for example lines 137-140: what is it the author(s) wants to convey here? The reasoning is not transparent, nor is the meaning of mothers “needing the most monitoring”? Are authors suggesting that mothers who are rejected e g due to lacking financial resources will not manage motherhood without “monitoring” by experts? If so, what is this assumption based upon?”

We rephrased this part in the hope it becomes more clear now: “Given the saying that it takes a village to raise a child, the evaluation whether an aspirant parent is able to raise a child is a complex decision. After all, the social conceptualization of good parenting is formed by the interaction of different social views (Morris et al., 2020). It is plausible that fertility practitioners are influenced by dominant ideas about parenthood in the screening procedure, but even so, there is a high probability that an ‘objective’ assessment leads to a rejection of singles with low financial and/or social capital because it is believed that this aggrevates the challenges of single parenthood (Whisenhunt et al., 2019). Regardless, rejected candidates can also become a parent via other means. , where they receives less formal support in the preparation of parenthood. This puts pressure on practitioners because rejection might does not withhold aspiring singles to become a parent. Hence, fertility practitioners face conflicting responsibilities: strict selection versus helping patients; or in other words: the interest of the mother versus the interest of the unborn child.”

  • “Ideas and concerns about risk for oneself as a professional, from making certain assessments and recommendations.”

In the current version of the manuscript, we elaborated more on the role change of practitioners in the literature section of the manuscript. As such, we now state: “Fertility gatekeeping defines norms and values of parenthood in the 21st century since it expresses dominant parenthood ideologies in a very applied manner (Sperling & Simon, 2010). (…) In contrast with those other areas, a parenthood ideology is explicitly voiced when single women are assessed on parental fitness. Fertility gatekeeping is an often unsolicited social role practitioners in fertility clinics have to take on. Fertility gatekeeping also means that not each and every patient can be treated, which is just often voiced as a main motivation for students to go into healthcare (Newton et al., 2009).” We hope this opens up more thinking about the concerns and risks of professionals to be a fertility gatekeeper but we like to hear it if we need to delve deeper into this in the results section.

  1. “Also, the assumption by (some) of the experts that they make “objective” assessments is highly interesting, and I suggest the author(s) critique/problematize this. See e g line 336, and compare to the quote just before, on lines 319-321. Could be interesting to bring this togeth”er with discussion of gatekeeper role and retreating from gatekeeper role on page 9.

We have followed through on this advice. See the second paragraph of section 5.1 Implications for parenting in the 21st century.

  1. “I would also add that the authors’ strong recommendation for “adjusted legislation” (page 10) based on the criteria list is a bit surprising, given that the interviewees do not raise further legislation as a felt need. Perhaps the argument for this could be built up to become more persuasive?”

We reworked this bit and structured a better argumentation, in line with other studies, supporting this claim for a more objective and legally embedded set of practical guidelines.

  • under Discussion – the 1st is repetitive, it is only in §2 (“The findings of this study” lines 384 ff) that we get to actual discussion, start here!

We agree with this comment and rewrote the discussion to start at §2.

Sentence structure: Lines 131-133: messy sentence, please revise.

We revised this sentence as follows: “Doctor Shopping and Unmonitored Fertilization might increase the complexity of the gatekeeping dilemma even further”

Word choices, e g Line 10 (abstract) result? Might mean “resort”?; Line 47 educate a child? Is this what parents are expected to do?

Thank you for pointing this out for us, we changed this.

Grammar/syntax: Line 125 the candidates’ rejection – sounds as if it is prospective mothers who are rejecting; please rephrase to convey that it is the experts who reject candidates.

We understand that this sentence was unclear, therefore, we rephrased: ““It grants clinical practitioners a high level of agency (Pennings, 2012) since there are few guidelines about the screening procedure and much reliance on a moral clause for the clinical practitioners to reject aspiring single mothers based on ethical considerations that would not meet the best interest of the child””

Prepositions: Line 7 (abstract) in account should be into account; Line 394 Aside for should be aside from.

Thank you for pointing this out, we adjusted both sentences. Thanks to both the comments of you, reviewer 1 and the editor, we have a feeling that the quality of the article is growing. We hope that our manuscript now better fits the expectations of you but are still open to revise and resubmit further concerns.

Kind regards

The Authors

Round 2

Reviewer 1 Report

I believe the manuscript has been sufficiently improved to warrant publication in Social Sciences.

Author Response

Dear reviewer

Thank you for your appreciation. We made some minor changes on demand of the other reviewer but want to thank you for your input in the process. 

Kind regards

The authors

Reviewer 2 Report

Enclosed is the article ms. with my comments. There are still a couple rough patches, and a couple statements that are not logical/not substantiated, these need some attention, but these are now much more minor changes than in the previous round of revisions. I hope these comments are helpful.

Author Response

Dear editor

Thank you for your detailed comments and remarks in the pdf version of the manuscript. We adjusted our manuscript using all of your suggestions. We also agree that one alinea was rather messy and solved this issue by splitting the alinea in two separate alineas. This way, we could formulate the main thoughts in this alinea more clearly. 

It was a pleasure to use your comments in reworking our manuscript and believe that this interaction improved our manuscript very much. 

Kind regards

the authors